# Determinants of women's empowerment in Nepal

**Daan-Max van Dongen[1], Maksym Obrizan[2]*, Vladyslav Shymanskyi[2]**

**1** Nuffield Department of Primary Care Health Sciences, The University of Oxford, Oxford, United Kingdom,
**2** Kyiv School of Economics, Kyiv, Ukraine

☯ These authors contributed equally to this work.
\* mobrizan@kse.org.ua

## Abstract

This study seeks to identify key determinants of women empowerment in Nepal using a rich set of socio-demographic and socio-economic characteristics as well as behavioral factors and regional indicators. Results showed that older age is generally associated with higher empowerment across all these domains, while partner controlling behavior tended to decrease empowerment in beliefs about violence and control over sexuality. Education level and wealth were correlated with increased empowerment in control over sexuality and safe sex, though not in the other two domains. Access to media showed mixed effects, reducing empowerment in decision-making but enhancing it in control over sexuality and safe sex. The results suggest that women's empowerment has morphed from the purview of gender equality programs to its current state, where it is considered a broader goal for development.

## Introduction

Nepal is characterized by low empowerment of women which may have negative effects on their health status as well as sexual and reproductive rights. We seek to identify key determinants of women empowerment in Nepal using a rich set of socio-demographic and socio-economic characteristics as well as behavioral factors and regional indicators.

## Methods

This study utilizes 4,211 women aged between 15 and 49 years from the 2022 Demographic and Health Survey (DHS) for Nepal. Following the previous study for Mozambique, we use Principal Component Analysis (PCA) to identify components of women's empowerment along three domains: beliefs about violence, decision-making and control over sexuality and safe sex. We use logistic regressions to identify significant predictors of empowerment in each domain and provide crude and adjusted odds ratios along with their 95% confidence intervals.

## Results

We found that older age is generally associated with higher empowerment across all these domains. Interestingly, while partner controlling behavior tended to decrease empowerment

**Data Availability Statement:** Data Availability Statement: The data used in this study are from the Demographic and Health Surveys (DHS) program, which are available for registered users at https://www.dhsprogram.com/methodology/survey/survey-display-585.cfm. These data are not owned

by the authors and cannot be made freely accessible due to restrictions imposed by the data provider. However, researchers can apply for access to the same data used in this study by registering and requesting permission from the DHS program (https://dhsprogram.com/data/new-user-registration.cfm). The authors did not have any special access privileges that others would not have. Upon approval, researchers can download the datasets used in this analysis from the DHS website.

**Funding:** MO gratefully acknowledges financial support from Kyiv School of Economics, Ukraine. The funders did not play any role in the study design, data collection and analysis, decision to publish, or preparation of the manuscript. All remaining errors are ours.

**Competing interests:** The authors have declared that no competing interests exist.

in beliefs about violence and control over sexuality, it was linked to increased decision-making empowerment. Notable regional differences emerged, with higher levels of empowerment observed in the Madhesh and Sudurpashchim regions. Further, education level and wealth were correlated with increased empowerment in control over sexuality and safe sex, though not in the other two domains. Access to media showed mixed effects, reducing empowerment in decision-making but enhancing it in control over sexuality and safe sex.

## Conclusion

Our results have many similarities but also notable differences with previous literature which emphasizes the importance of regular and region-specific studies of women's empowerment, acknowledging the potential for its change over time and also the prevailing differences across regions.

## Introduction

Women empowerment has become a widely recognised global policy objective and a vital component of the diverse raft of measures premised on improving the overall life outcomes of populations around the globe. The relevance of the discourse on women's empowerment on the global scene is reflected by the initiatives being pushed by different international organisations, such as the United Nations, to fast-track women's emancipation. Of particular relevance to that end are the Sustainable Development Goals (SDG), particularly SDG 5, launched in 2015, which calls for a global commitment towards the eradication of gender inequality, as well as promoting the empowerment of women and girls [1]. As a relatively new concept, the different studies on empowerment have yielded varied definitions. Even so, empowerment can be generally understood as the capacity to exercise choice and freedom to make decisions, which had previously been denied [2].

Generally, empowerment emanates from the confluence of two critical concepts, including education and income preconditions. The second concept is agency, which is the capacity to choose and make decisions for oneself [2]. Thus, women's empowerment is intricately tied to their capacity to decide for themselves by affording them the power and control over their lives, thus making it possible for them to make strategic decisions [3]. Over time, women's empowerment has morphed from the purview of gender equality programs to its current state, where it is considered a broader goal for development [4]. However, as with other socio-economic and political indicators of societies around the globe, there are significant disparities in the level of empowerment of women in different regions around the globe [3]. Specifically, in the more developed nations, the prospects of women appear considerably better than for most women in the less developed nations, particularly in the global south.

The disparities in women's empowerment globally can be explained by studies which highlight that aside from education and income, other socio-economic factors such as age to marriage, religion, and access to land, all tally up towards determining a woman's level of empowerment [2]. Furthermore, gaining support in other regions globally has taken time because women's empowerment is perceived mainly as a Western ideology, having emerged from more developed nations such as the United States. Thus, as evidence would suggest, gender discrimination tends to be more severe in the developing world, and that reality is well reflected in Nepal [3]. That reality is reflected in the low scores on the various women

empowerment indicators such as the human development index (HDI), gender development index (GDI), gender inequality index (GII) and the global gender gap index (GGGI) [3]. According to World Bank data, Nepal ranks as one of the poorest nations globally, with a gross national per capita income of $960 and $3,090 based on the purchasing power parity indicator [5]. Furthermore,, with a HDI of 0.574, Nepal is ranked 149 out of 189 countries and ranks 118th globally on GII with a score of 0.480 [5].

Some of the various population indicators that would explain the poor life prospects of many Nepalese is that 15 per cent of the population lives on less than $1.90 daily. In comparison, 32.6 per cent of the employed population falls into the working poor category, meaning they live on less than $3.10 daily [5]. Furthermore, as a testament to its more traditional economic base, only 20 per cent of the country's population resides in urban areas, with over 70 per cent of the employed being engaged in the agricultural sector [5].

In addition, Nepal ranks poorly on many socio-economic indicators, especially for women who may face additional discriminatory tendencies. This is confirmed, for example, by high maternal mortality ratio, estimated at 151 per 100,000 live births, with the Lumbini and Karnali provinces registering figures as high as 207 deaths for 100,000 live births [6]. Hence, society must embrace women's empowerment in order to improve their life outcomes given the current low performance on many socio-economic indicators for women. It is our goal in this study to evaluate the current state of women empowerment in Nepal, using recent 2022 DHS survey.

The rest of the paper is organized as follows. First, we discuss in detail the Demographic and Health Survey (DHS-VIII) conducted in Nepal in 2022 [7] which we used in this study. Then we define the list of empowerment indicators and independent variables that are utilized to model women empowerment. Next, we provide details about the methodology and results. Finally, we discuss the findings, compare them to the previous literature, provide the list of limitations and conclude in the last two sections.

## Data source

This study uses the most recent Demographic and Health Survey (DHS-VIII) conducted in Nepal in 2022 that spanned the entire country, divided into seven regions. The DHS program, supported by the United States Agency for International Development (USAID), aids countries in evaluating their demographic and health statistics comprehensively at both national and regional levels. The DHS Program ensures that all collected data comply with ethical standards, including informed consent from participants, and the data is anonymized to protect individuals' privacy [7].

The survey includes detailed socio-economic and demographic information about the participants over a range of topics, such as infant vaccination rates, malaria prevalence, knowledge and testing for HIV, fertility rates, family planning, prenatal care, insights into women's empowerment, and domestic violence reports [7]. This approach offers a broad perspective on crucial topics related to population, health, and nutrition.

The DHS in Nepal used a rigorous sampling process designed to ensure that the survey's findings are genuinely representative at both national and regional levels. Preceding the primary survey, the questionnaire underwent thorough pretesting in non-selected areas across the country to fine-tune and enhance the clarity and relevance of the questions. Interviewers underwent a comprehensive training program encompassing both theoretical and practical aspects to guarantee the precision and consistency of data collection.

Data was collected from all individuals aged 15 to 49, irrespective of whether they were residents or overnight visitors in the selected households. Interviews were completed with 14,845

women aged 15–49 yielding a response rate of 97% from 15,238 women who were identified as eligible for individual interviews [8].

The sections related to women's empowerment and issues surrounding violence were selectively administered to women who were either married or in a union. After cleaning the observations with missing data, a total of 4,211 women (or 85.7%) were included in this in-depth analysis.

## Outcome variables

### Empowerment indicators

This study focuses on three essential domains of women's empowerment following an earlier study based on DHS survey in Mozambique [2]: beliefs about violence against women, decision-making and control over sexuality and sex. Empowerment is defined as the ability to control and make free decisions concerning one's life and body, ultimately achieving valued or personally determined outcomes.

In this study, we focus on three key areas of women's empowerment. First, we look at beliefs about violence against women to gain insight into how societal norms and personal views influence their sense of empowerment. Second, we assess women's participation in decision-making, evaluating their involvement in choices regarding healthcare, household expenditures, and social activities. Third, we explore women's control over their sexuality and sex, examining their autonomy in managing their sexual health and their ability to negotiate safe sex practices, such as the use of condoms to prevent sexually transmitted infections. The investigation involves understanding the factors that influence empowerment within this context, aiming to enhance women's sexual health rights.

We apply a 3-point scale (-1, 0, 1) to the selected empowerment indicators, aligning with a methodology applied to a 2015 DHS survey for Mozambique [2], wherein higher values signify greater empowerment. This approach allows us to distinguish between fully empowered, partially empowered, and disempowered individuals. Table 1 below provides further details about women empowerment in Nepal based on the sample from 2022 DHS survey used in this study.

### Independent variables

Utilizing data from the Nepal DHS, we include in the analysis a range of socioeconomic, demographic, and behavioral indicators as independent variables.

Socioeconomic and Demographic Variables:

1. Age Groups: Age categories divide individuals aged "Less or equal to 19," "20–29," "30–39," and "40–49 years.". Hereinafter we use the same definition of variables (i.e. using age groups instead of a continuous age) for a better comparability of results for Nepal with the earlier study based on the DHS survey for Mozambique [2].

2. Education Levels: Participants' education status was grouped into "No Education", "Incomplete Primary", "Complete Primary", "Incomplete Secondary", "Complete Secondary", and "Higher"

3. Employment Status: Employment situations were categorized as either "Working" or "Not working."

4. Age at First Cohabitation: This variable included age brackets of "10 to 14," "15 to 19," and "20 and above years."

**Table 1. Indicators of women's empowerment by 7 regions in Nepal.**

| | Koshi | Madhesh | Bagmati | Gandaki | Lumbini | Karnali | Sudurpashchim | Total |
|---|---|---|---|---|---|---|---|---|
| | (N = 645) | (N = 709) | (N = 588) | (N = 504) | (N = 620) | (N = 580) | (N = 565) | (N = 4211) |
| | | | | | Who usually decides on: | | | |
| *Woman's health care* | | | | | | | | |
| Woman Alone | 148 (22.9%) | 121 (17.1%) | 149 (25.3%) | 127 (25.2%) | 142 (22.9%) | 106 (18.3%) | 108 (19.1%) | 901 (21.4%) |
| Jointly | 312 (48.4%) | 293 (41.3%) | 305 (51.9%) | 287 (56.9%) | 319 (51.5%) | 321 (55.3%) | 338 (59.8%) | 2175 (51.7%) |
| Partner Or Other Alone | 185 (28.7%) | 295 (41.6%) | 134 (22.8%) | 90 (17.9%) | 159 (25.6%) | 153 (26.4%) | 119 (21.1%) | 1135 (27.0%) |
| *Large household purchases* | | | | | | | | |
| Woman Alone | 185 (28.7%) | 121 (17.1%) | 157 (26.7%) | 188 (37.3%) | 159 (25.6%) | 136 (23.4%) | 114 (20.2%) | 1060 (25.2%) |
| Jointly | 214 (33.2%) | 256 (36.1%) | 237 (40.3%) | 177 (35.1%) | 215 (34.7%) | 254 (43.8%) | 222 (39.3%) | 1575 (37.4%) |
| Partner Or Other Alone | 246 (38.1%) | 332 (46.8%) | 194 (33.0%) | 139 (27.6%) | 246 (39.7%) | 190 (32.8%) | 229 (40.5%) | 1576 (37.4%) |
| *Visits to family or relatives* | | | | | | | | |
| Woman Alone | 247 (38.3%) | 134 (18.9%) | 212 (36.1%) | 173 (34.3%) | 180 (29.0%) | 141 (24.3%) | 157 (27.8%) | 1244 (29.5%) |
| Jointly | 254 (39.4%) | 263 (37.1%) | 253 (43.0%) | 225 (44.6%) | 236 (38.1%) | 275 (47.4%) | 228 (40.4%) | 1734 (41.2%) |
| Partner Or Other Alone | 144 (22.3%) | 312 (44.0%) | 123 (20.9%) | 106 (21.0%) | 204 (32.9%) | 164 (28.3%) | 180 (31.9%) | 1233 (29.3%) |
| | | | | | Beating justified if: | | | |
| *Wife goes out without telling husband* | | | | | | | | |
| Not Justified | 601 (93.2%) | 670 (94.5%) | 541 (92.0%) | 469 (93.1%) | 573 (92.4%) | 523 (90.2%) | 538 (95.2%) | 3915 (93.0%) |
| Don't Know | 0 (0.0%) | 2 (0.3%) | 1 (0.2%) | 0 (0.0%) | 1 (0.2%) | 2 (0.3%) | 0 (0.0%) | 6 (0.1%) |
| Justified | 44 (6.8%) | 37 (5.2%) | 46 (7.8%) | 35 (6.9%) | 46 (7.4%) | 55 (9.5%) | 27 (4.8%) | 290 (6.9%) |
| *Wife neglects the children* | | | | | | | | |
| Not Justified | 535 (82.9%) | 647 (91.3%) | 492 (83.7%) | 432 (85.7%) | 515 (83.1%) | 461 (79.5%) | 479 (84.8%) | 3561 (84.6%) |
| Don't Know | 1 (0.2%) | 3 (0.4%) | 0 (0.0%) | 0 (0.0%) | 1 (0.2%) | 0 (0.0%) | 0 (0.0%) | 5 (0.1%) |
| Justified | 109 (16.9%) | 59 (8.3%) | 96 (16.3%) | 72 (14.3%) | 104 (16.8%) | 119 (20.5%) | 86 (15.2%) | 645 (15.3%) |
| *Wife argues with husband* | | | | | | | | |
| Not Justified | 606 (94.0%) | 651 (91.8%) | 558 (94.9%) | 490 (97.2%) | 589 (95.0%) | 543 (93.6%) | 540 (95.6%) | 3977 (94.4%) |
| Don't Know | 2 (0.3%) | 1 (0.1%) | 1 (0.2%) | 0 (0.0%) | 0 (0.0%) | 1 (0.2%) | 0 (0.0%) | 5 (0.1%) |
| Justified | 37 (5.7%) | 57 (8.0%) | 29 (4.9%) | 14 (2.8%) | 31 (5.0%) | 36 (6.2%) | 25 (4.4%) | 229 (5.4%) |
| *Wife refuses to have sex with husband* | | | | | | | | |
| Not Justified | 637 (98.8%) | 699 (98.6%) | 572 (97.3%) | 495 (98.2%) | 602 (97.1%) | 553 (95.3%) | 558 (98.8%) | 4116 (97.7%) |
| Don't Know | 0 (0.0%) | 2 (0.3%) | 4 (0.7%) | 2 (0.4%) | 0 (0.0%) | 1 (0.2%) | 0 (0.0%) | 9 (0.2%) |
| Justified | 8 (1.2%) | 8 (1.1%) | 12 (2.0%) | 7 (1.4%) | 18 (2.9%) | 26 (4.5%) | 7 (1.2%) | 86 (2.0%) |
| *Wife burns the food* | | | | | | | | |
| Not Justified | 631 (97.8%) | 691 (97.5%) | 578 (98.3%) | 499 (99.0%) | 614 (99.0%) | 569 (98.1%) | 562 (99.5%) | 4144 (98.4%) |
| Don't Know | 3 (0.5%) | 0 (0.0%) | 2 (0.3%) | 0 (0.0%) | 1 (0.2%) | 2 (0.3%) | 1 (0.2%) | 9 (0.2%) |
| Justified | 11 (1.7%) | 18 (2.5%) | 8 (1.4%) | 5 (1.0%) | 5 (0.8%) | 9 (1.6%) | 2 (0.4%) | 58 (1.4%) |
| | | | | | A woman can: | | | |
| *Ask husband to use condom if he has STI* | | | | | | | | |
| Yes | 609 (94.4%) | 634 (89.4%) | 520 (88.4%) | 458 (90.9%) | 569 (91.8%) | 501 (86.4%) | 535 (94.7%) | 3826 (90.9%) |
| No | 25 (3.9%) | 29 (4.1%) | 44 (7.5%) | 36 (7.1%) | 30 (4.8%) | 57 (9.8%) | 17 (3.0%) | 238 (5.7%) |
| Don't Know | 11 (1.7%) | 46 (6.5%) | 24 (4.1%) | 10 (2.0%) | 21 (3.4%) | 22 (3.8%) | 13 (2.3%) | 147 (3.5%) |
| *Refuse sex* | | | | | | | | |
| Yes | 614 (95.2%) | 595 (83.9%) | 554 (94.2%) | 467 (92.7%) | 577 (93.1%) | 527 (90.9%) | 533 (94.3%) | 3867 (91.8%) |
| No | 30 (4.7%) | 112 (15.8%) | 33 (5.6%) | 35 (6.9%) | 41 (6.6%) | 53 (9.1%) | 30 (5.3%) | 334 (7.9%) |
| Don't Know | 1 (0.2%) | 2 (0.3%) | 1 (0.2%) | 2 (0.4%) | 2 (0.3%) | 0 (0.0%) | 2 (0.4%) | 10 (0.2%) |
| *Ask partner to use a condom* | | | | | | | | |
| Yes | 558 (86.5%) | 386 (54.4%) | 502 (85.4%) | 427 (84.7%) | 516 (83.2%) | 473 (81.6%) | 508 (89.9%) | 3370 (80.0%) |
| No | 75 (11.6%) | 314 (44.3%) | 75 (12.8%) | 66 (13.1%) | 101 (16.3%) | 105 (18.1%) | 52 (9.2%) | 788 (18.7%) |
| Don't Know | 12 (1.9%) | 9 (1.3%) | 11 (1.9%) | 11 (2.2%) | 3 (0.5%) | 2 (0.3%) | 5 (0.9%) | 53 (1.3%) |

5. Marital Structure: Marital status was examined through categories such as "Not polyga-mous," "Polygamous," and "Doesn't know."

6. Religious Affiliation: Participants' religious affiliations were classified as "Hindu," "Bud-dhist," "Muslim," "Kirat," "Christian," or "Other."

Geographic Factors:

The study included indicators for Nepal's seven provinces (Koshi, Madhesh, Bagmati, Gan-daki, Lumbini, Karnali, and Sudurpashchim) as well as participants' urban or rural area of residency.

Wealth Index Quintiles:

The wealth index, computed by the DHS, combined various factors like ownership of televi-sions, bicycles, housing construction materials, water access, and sanitation facilities. This index was further divided into quintiles: "Poorest," "Poor," "Middle," "Rich," and "Richest."

Media Access Indicator:

An indicator denoting access to media was created based on participants' responses regard-ing the frequency of reading newspapers or magazines, listening to the radio, and watching TV. This variable was assigned values: "0" for "Not at all," "1" for "Less than once a week," and "2" for "At least once a week."

Control Behaviors and Intimate Partner Violence (IPV):

The study considered women's exposure to controlling behaviors and IPV. For controlling behaviors, a binary variable was created, distinguishing between "No control" and "At least one type of control." IPV was assessed using three variables to determine if women had ever experi-enced any form of violence from their husbands/partners.

## Methodology

To better understand the links between various socioeconomic, demographic, behavioral indi-cators, and those of women's empowerment, we applied a number of methods identified in the previous literature on women's empowerment [2]. All data analyses were conducted using the statistical packages in R software.

For descriptive statistics we used cross-tabulation and chi-squared tests to compare propor-tions across different regions.

We conducted a Principal Component Analysis (PCA) to recast the multi-indicator dataset into a set of uncorrelated variables. This technique aimed to retain most of the variance from the original data. When applying PCA we used ordinal encoding for the empowerment indexes [2], coding them into a 3-point scale (i.e., values of -1, 0, 1) where the highest value was assigned to categories representing greater levels of empowerment. In particular, we used variables from Table 1 in the PCA to generate the empowerment indexes.

Utilizing the results from the scree plot of the PCA, significant components (eigenvalues greater than 1) were retained. Subsequently, an orthogonal varimax rotation was applied, ensuring that there were no correlations between the retained components [9]. This methodol-ogy was applied to achieve greater objectivity in evaluating the patterns of empowerment indi-cators and their respective contributions, as previously implemented in studies on women's empowerment [2]. In addition, the Kaiser Meyer-Olkin (KMO) measure of sampling adequacy was employed to assess the suitability of the data for PCA, ensuring robust results and inferences.

Domain-specific empowerment indexes were derived using the PCA factor scores. These indexes were further categorized into quintiles, ranging from the most empowered women (5th quintile) to the least empowered (1st quintile). For analytical purposes, women were

classified into two categories: the most empowered (5th quintile) and the less empowered (all groups below the 5th quintile).

Logistic regression was employed to estimate the associations between socioeconomic, demographic, and behavioral characteristics and empowerment within each domain. The models were both unadjusted and adjusted to account for the influence of women's education. This adjustment aimed to investigate whether the relationships found between the selected characteristics and empowerment were independent of women's educational backgrounds. The inclusion of different characteristics in the final models was guided by both theoretical underpinnings and statistical significance (with a predetermined threshold of 0.05). Crude and adjusted odds ratios (OR) were calculated to evaluate the associations, with respective 95% confidence intervals (95% CI) to provide a range for the estimates.

The fit of empowerment domains across regions was explored through interaction term/ tests between each domain and region. However, no significant differences were detected, leading to the presentation of results that encompass all regions. The variable region was retained in the final model as a control variable.

## Results

In Nepal there exist important provincial differences in terms of geography, population, economy and so on. Bagmati province, for example, includes the Kathmandu Valley and is the main powerhouse of the economy. In addition, Bagmati is the largest province in terms of population share (20.6%) compared to the smallest province of Sudurpashchim (only 8.6% of the population) [7].

7 Nepal provinces also differ in terms of political representation measured by Federal Electoral constituency ranked as: Karnali (12), Sudurpashchim (16), Gandaki (18), Lumbini (26), Koshi (28), Madhesh (32), Bagmati (33) [10].

Provinces also differ in terms of the poverty rates [11]. The province of Sudurpashchim has the highest poverty rate (34.2% poverty rate), followed by the province of Karnali (26.7%). Bagmati and Gandaki, on the other hand, have the lowest poverty rates of 12.6% and 11.9% correspondingly. The remaining three provinces are in the middle in terms of their poverty rates: Koshi (17.2%), Madhesh (22.5%) and Lumbini (24.4%).

Table 1 further illustrates the empowerment indicators among women which confirms significant differences between the regions outlined above.

The Koshi region showed significant differences when compared to the Madhesh region. In Koshi, more women reported making large household purchases and deciding on visits to family or relatives alone (p-value < 0.001). Additionally, women in Koshi reported a higher ability to ask their partners to use condoms if they have an STI (p-value < 0.001). A high percentage of women in Koshi felt that beating is not justified in any situation (p-value < 0.001).

The Madhesh region exhibited significant differences from Koshi, especially in the category of large household purchases (p-value = 0.031). Women in Madhesh reported more decision-making autonomy in this aspect. They also expressed a strong stance against justification for beating in any situation (p-value < 0.001).

The Bagmati region had variations compared to Koshi and Madhesh, particularly in the category of visits to family or relatives, where more women in Bagmati reported making these decisions alone (p-value = 0.047). Women from Bagmati generally held strong views against justifying beating in any situation, but there were no significant regional differences.

The Gandaki region displayed differences when compared to Madhesh, especially regarding women making decisions about visits to family or relatives (p-value = 0.026). Women in

Gandaki were also adamant about not justifying beating, with no significant regional differences (p-value < 0.001).

In Lumbini, differences were observed compared to Koshi and Madhesh, notably in the empowerment indicator of women being able to ask their partners to use condoms if they have an STI (p-value < 0.001). Lumbini reported the highest percentage of women justifying beating when the woman refuses to have sex (p-value = 0.005).

Karnali showed significant differences compared to Koshi, Madhesh, and Bagmati, particularly in women's ability to ask their partners to use condoms if they have an STI (p-value < 0.001). Additionally, women in Karnali expressed significant regional difference in justifying beating when the woman refuses to have sex (p-value = 0.010).

Sudurpashchim differed from Koshi and Madhesh in various aspects. More women in Sudurpashchim reported the ability to ask their partners to use condoms (p-value < 0.001) and were more likely to make large household purchases and decisions about visits to family or relatives alone. Women in Sudurpashchim had a strong stance against justification of beating in any situation (p-value < 0.001).

The socio-demographic and socio-economic characteristics of the women included in the final sample are presented in Table 2. The data, comprising age distribution, education, wealth index, religion, urban vs. rural residency, age of first cohabitation, polygamous marriage, employment status, partner controlling behavior, IPV exposure (Intimate Partner Violence), and access to media, provides crucial insights into the diverse characteristics and socio-cultural dynamics of these regions.

Most respondents belong to age groups of 25–29 (20.8% of the sample) and 30–34 (19.9% of the sample), while women of 15–19 represent only 4.3% of all observations. The variation in age groups across regions is relatively small compared to the level of education. For example, the "No Education" category includes only 20.9% of women in Koshi region and as much as 52.5% of females in Madhesh region.

Economic stratification, as reflected by the wealth index, is characterized by the "Poorest" category, reaching maximum in Karnali at 69.3% and minimum in Lumbini at 17.6%. In contrast, "Richest" category displays an inverse trend, with Karnali region recording the lowest percentage (3.1%) and Bagmati region the highest (29.9%) respectively. Religious affiliation is primarily Hindu across regions, spanning from 65.6% in Koshi region to 96.8% in Sudurpashchim region, while Buddhist and Muslim populations have bigger representation in specific regions, notably Bagmati (20.6%) and Madhesh (11.4%) respectively.

Regional variations also emerge in urban vs. rural residency, age of first cohabitation, and marital practices such as polygamy. The largest share of urban respondents has been interviewed in Madhesh region (55.3%) while the largest share of rural residents was surveyed in Karnali region (52.9%). The most common age for the first cohabitation is from 15 to 19 years old and varies from 52.0% in Bagmati region to 73.5% in Madhesh region. There is some variation in the share of respondents with the first cohabitation from 10 to 14 years old—from 6.5% in the Koshi region to as much as 16.4% in the Madhesh region. Polygamous marriages are in general not very wide-spread and include from the minimum of 1.6% in Karnali region to a maximum of 3.2% in Sudurpashchim region.

Furthermore, employment status showcases regional distinctions, with "Currently Employed" individuals being most widely represented in Gandaki region (79.8%) and least so in Lumbini region (66.5%).

Partner controlling behavior, indicating power dynamics within relationships, is most significant in Madhesh region (46.7%) and least in Sudurpashchim (20.0%). The presence of IPV exposure demonstrates regional variations, with Madhesh reporting the highest prevalence (23.1%) and Sudurpashchim the lowest (6.9%). Lastly, access to media, a critical channel for

**Table 2. Socioeconomic, demographic and behavioural characteristics of women included in the study.**

| | Koshi | Madhesh | Bagmati | Gandaki | Lumbini | Karnali | Sudurpashchim | Total |
|---|---|---|---|---|---|---|---|---|
| | (N = 645) | (N = 709) | (N = 588) | (N = 504) | (N = 620) | (N = 580) | (N = 565) | (N = 4211) |
| **Age (years)** | | | | | | | | |
| 15–19 | 27 (4.2%) | 44 (6.2%) | 11 (1.9%) | 16 (3.2%) | 23 (3.7%) | 38 (6.6%) | 24 (4.2%) | 183 (4.3%) |
| 20–24 | 100 (15.5%) | 125 (17.6%) | 56 (9.5%) | 61 (12.1%) | 99 (16.0%) | 111 (19.1%) | 86 (15.2%) | 638 (15.2%) |
| 25–29 | 123 (19.1%) | 142 (20.0%) | 121 (20.6%) | 93 (18.5%) | 148 (23.9%) | 120 (20.7%) | 127 (22.5%) | 874 (20.8%) |
| 30–34 | 126 (19.5%) | 133 (18.8%) | 128 (21.8%) | 116 (23.0%) | 118 (19.0%) | 105 (18.1%) | 112 (19.8%) | 838 (19.9%) |
| 35–39 | 115 (17.8%) | 113 (15.9%) | 117 (19.9%) | 84 (16.7%) | 102 (16.5%) | 87 (15.0%) | 85 (15.0%) | 703 (16.7%) |
| 40–44 | 88 (13.6%) | 96 (13.5%) | 86 (14.6%) | 61 (12.1%) | 80 (12.9%) | 67 (11.6%) | 65 (11.5%) | 543 (12.9%) |
| 45–49 | 66 (10.2%) | 56 (7.9%) | 69 (11.7%) | 73 (14.5%) | 50 (8.1%) | 52 (9.0%) | 66 (11.7%) | 432 (10.3%) |
| **Education** | | | | | | | | |
| No Education | 135 (20.9%) | 372 (52.5%) | 162 (27.6%) | 92 (18.3%) | 157 (25.3%) | 202 (34.8%) | 225 (39.8%) | 1345 (31.9%) |
| Incomplete Primary | 203 (31.5%) | 159 (22.4%) | 180 (30.6%) | 181 (35.9%) | 214 (34.5%) | 155 (26.7%) | 134 (23.7%) | 1226 (29.1%) |
| Complete Primary | 56 (8.7%) | 25 (3.5%) | 29 (4.9%) | 36 (7.1%) | 39 (6.3%) | 41 (7.1%) | 28 (5.0%) | 254 (6.0%) |
| Incomplete Secondary | 202 (31.3%) | 98 (13.8%) | 111 (18.9%) | 116 (23.0%) | 134 (21.6%) | 106 (18.3%) | 107 (18.9%) | 874 (20.8%) |
| Complete Secondary | 39 (6.0%) | 42 (5.9%) | 67 (11.4%) | 55 (10.9%) | 59 (9.5%) | 72 (12.4%) | 59 (10.4%) | 393 (9.3%) |
| Higher | 10 (1.6%) | 13 (1.8%) | 39 (6.6%) | 24 (4.8%) | 17 (2.7%) | 4 (0.7%) | 12 (2.1%) | 119 (2.8%) |
| **Wealth index** | | | | | | | | |
| Poorest | 181 (28.1%) | 52 (7.3%) | 118 (20.1%) | 103 (20.4%) | 109 (17.6%) | 402 (69.3%) | 271 (48.0%) | 1236 (29.4%) |
| Poorer | 126 (19.5%) | 204 (28.8%) | 104 (17.7%) | 111 (22.0%) | 134 (21.6%) | 79 (13.6%) | 105 (18.6%) | 863 (20.5%) |
| Middle | 133 (20.6%) | 222 (31.3%) | 98 (16.7%) | 107 (21.2%) | 149 (24.0%) | 38 (6.6%) | 79 (14.0%) | 826 (19.6%) |
| Richer | 140 (21.7%) | 162 (22.8%) | 92 (15.6%) | 109 (21.6*%) | 137 (22.1%) | 43 (7.4%) | 65 (11.5%) | 748 (17.8%) |
| Richest | 65 (10.1%) | 69 (9.7%) | 176 (29.9%) | 74 (14.7%) | 91 (14.7%) | 18 (3.1%) | 45 (8.0%) | 538 (12.8%) |
| **Religion** | | | | | | | | |
| Hindu | 423 (65.6%) | 615 (86.7%) | 413 (70.2%) | 443 (87.9%) | 573 (92.4%) | 547 (94.3%) | 547 (96.8%) | 3561 (84.6%) |
| Buddhist | 63 (9.8%) | 12 (1.7%) | 121 (20.6%) | 42 (8.3%) | 6 (1.0%) | 13 (2.2%) | 3 (0.5%) | 260 (6.2%) |
| Muslim | 24 (3.7%) | 81 (11.4%) | 3 (0.5%) | 2 (0.4%) | 20 (3.2%) | 1 (0.2%) | 0 (0.0%) | 131 (3.1%) |
| Kirat | 115 (17.8%) | 1 (0.1%) | 8 (1.4%) | 1 (0.2%) | 0 (0.0%) | 0 (0.0%) | 0 (0.0%) | 125 (3.0%) |
| Christian | 18 (2.8%) | 0 (0.0%) | 43 (7.3%) | 16 (3.2%) | 21 (3.4%) | 19 (3.3%) | 15 (2.7%) | 132 (3.1%) |
| Other | 2 (0.3%) | 0 (0.0%) | 0 (0.0%) | 0 (0.0%) | 0 (0.0%) | 0 (0.0%) | 0 (0.0%) | 2 (0.0%) |
| **Urban vs rural residency** | | | | | | | | |
| Urban | 334 (51.8%) | 392 (55.3%) | 318 (54.1%) | 251 (49.8%) | 321 (51.8%) | 273 (47.1%) | 292 (51.7%) | 2181 (51.8%) |
| Rural | 311 (48.2%) | 317 (44.7%) | 270 (45.9%) | 253 (50.2%) | 299 (48.2%) | 307 (52.9%) | 273 (48.3%) | 2030 (48.2%) |
| **Age of first cohabitation (years)** | | | | | | | | |
| Up To 10 | 1 (0.2%) | 0 (0.0%) | 4 (0.7%) | 0 (0.0%) | 0 (0.0%) | 1 (0.2%) | 3 (0.5%) | 9 (0.2%) |
| 10 To 14 | 42 (6.5%) | 116 (16.4%) | 41 (7.0%) | 45 (8.9%) | 56 (9.0%) | 72 (12.4%) | 67 (11.9%) | 439 (10.4%) |
| 15 To 19 | 361 (56.0%) | 521 (73.5%) | 306 (52.0%) | 290 (57.5%) | 395 (63.7%) | 397 (68.4%) | 368 (65.1%) | 2638 (62.6%) |
| 20 Or Above | 241 (37.4%) | 72 (10.2%) | 237 (40.3%) | 169 (33.5%) | 169 (27.3%) | 110 (19.0%) | 127 (22.5%) | 1125 (26.7%) |
| **Polygamous marriage** | | | | | | | | |
| No Polygamous | 629 (97.5%) | 686 (96.8%) | 568 (96.6%) | 493 (97.8%) | 604 (97.4%) | 570 (98.3%) | 545 (96.5%) | 4095 (97.2%) |
| Polygamous | 11 (1.7%) | 13 (1.8%) | 18 (3.1%) | 9 (1.8%) | 11 (1.8%) | 9 (1.6%) | 18 (3.2%) | 89 (2.1%) |
| Does Not Know | 5 (0.8%) | 10 (1.4%) | 2 (0.3%) | 2 (0.4%) | 5 (0.8%) | 1 (0.2%) | 2 (0.4%) | 27 (0.6%) |
| **Currently employed** | | | | | | | | |
| No | 202 (31.3%) | 335 (47.2%) | 135 (23.0%) | 102 (20.2%) | 208 (33.5%) | 118 (20.3%) | 181 (32.0%) | 1281 (30.4%) |
| Yes | 443 (68.7%) | 374 (52.8%) | 453 (77.0%) | 402 (79.8%) | 412 (66.5%) | 462 (79.7%) | 384 (68.0%) | 2930 (69.6%) |
| **Partner controlling behavior** | | | | | | | | |
| No | 434 (67.3%) | 378 (53.3%) | 436 (74.1%) | 369 (73.2%) | 386 (62.3%) | 409 (70.5%) | 452 (80.0%) | 2864 (68.0%) |
| At Least One Type | 211 (32.7%) | 331 (46.7%) | 152 (25.9%) | 135 (26.8%) | 234 (37.7%) | 171 (29.5%) | 113 (20.0%) | 1347 (32.0%) |

*(Continued)*

**Table 2.** (Continued)

| | Koshi | Madhesh | Bagmati | Gandaki | Lumbini | Karnali | Sudurpashchim | Total |
|---|---|---|---|---|---|---|---|---|
| | (N = 645) | (N = 709) | (N = 588) | (N = 504) | (N = 620) | (N = 580) | (N = 565) | (N = 4211) |
| **IPV exposure** | | | | | | | | |
| No | 585 (90.7%) | 545 (76.9%) | 544 (92.5%) | 461 (91.5%) | 542 (87.4%) | 536 (92.4%) | 526 (93.1%) | 3739 (88.8%) |
| Yes | 60 (9.3%) | 164 (23.1%) | 44 (7.5%) | 43 (8.5%) | 78 (12.6%) | 44 (7.6%) | 39 (6.9%) | 472 (11.2%) |
| **Access to media** | | | | | | | | |
| Not At All | 107 (16.6%) | 264 (37.2%) | 81 (13.8%) | 90 (17.9%) | 133 (21.5%) | 152 (26.2%) | 87 (15.4%) | 914 (21.7%) |
| Less Than Once A Week | 334 (51.8%) | 275 (38.8%) | 285 (48.5%) | 240 (47.6%) | 248 (40.0%) | 233 (40.2%) | 323 (57.2%) | 1938 (46.0%) |
| At Least Once A Week | 204 (31.6%) | 170 (24.0%) | 222 (37.8%) | 174 (34.5%) | 239 (38.5%) | 195 (33.6%) | 155 (27.4%) | 1359 (32.3%) |

information dissemination, reveals noteworthy differences, with "At Least Once A Week" access to media most common in Lumbini region (38.5%) and least common in Madhesh region (24.0%).

With the PCA three significant components were retained. The three retained components explained 20%, 17%, and 14% of the total variance, respectively, adding up to 51%. The KMO test value was 0.68 therefore we consider the sampling adequate for PCA. The retained components were then identified as empowerment domains and included: Beliefs about violence against women; Decision-making; and Control over sexuality and safe sex following the previous literature [2].

The results of the final regression for the association between the socio-economic, demographic, and behavioral characteristics and the different domains of empowerment are provided in Table 3. Older age is associated with higher empowerment in the domain of beliefs about violence against women with odds ratios adjusted for education (aORs) taking values from 1.71 to 2.08 (significant at 5%) but without a clear correlation with age. Women living in Madhesh and Sudurpashchim regions also demonstrate higher empowerment with corresponding odds ratios of 1.58 and 1.73 (significant at 5%). At least one type of partner controlling behavior is the only significant factor reducing empowerment in the domain of beliefs about violence with an aOR of 0.55.

When it comes to the domain of decision-making older age is also associated with higher odds of women's empowerment with adjusted odds ratios of 2.29 to a very high of 10.08 (significant at 5%). Somewhat surprisingly, polygamous marriage (aOR of 2.98) and partner control behaviour (aOR of 1.33) are also associated with increased odds of being in the top quintile in terms of empowerment. Also, having some access to media is associated with reduced empowerment (aOR of 0.63–0.67) relative to now access to media.

The largest number of significant factors has been identified in the domain of control over sexuality and safe sex. Here again older age is associated with higher chances of empowerment with adjusted odds ratios of 1.52 to 2.03 (significant at 5%). Some education also provides higher changes of being empowered compared to no education with adjusted odds ratios of 1.52 to 6.38 but without clear correlation with years of education. There are also some regional differences (with higher empowerment in regions of Koshi and Sudurpashchim and lower in the region of Madhesh) and in terms of religion. Higher wealth is associated with increased empowerment (aOR of 1.29–1.91) as well as having some access to media (aOR of 1.72–1.45) relative to now access to media. Finally, partner control behaviour (aOR of 0.81) is also associated with reduced odds of being in the top quintile in terms of empowerment in the domain of control over sexuality and safe sex.

**Table 3. Crude and adjusted odds ratios for the association between the socio-economic, demographic, and behavioral characteristics and the different domains of empowerment.**

| | Beliefs about violence against women | | Decision-making | | Control over sexuality and sex | |
|---|---|---|---|---|---|---|
| | cOR (95% CI) | aOR (95% CI) | cOR (95% CI) | aOR (95% CI) | cOR (95% CI) | aOR (95% CI) |
| **Age** | | | | | | |
| 20–24 | 1.96*** | 1.92*** | 2.25 | 2.29 | 1.39* | 1.37 |
| | (1.19, 3.19) | (1.16, 3.12) | (0.76, 9.63) | (0.77, 9.81) | (0.96, 2.03) | (0.93, 2) |
| 25–29 | 2.10*** | 2.08*** | 5.39*** | 5.58*** | 1.82*** | 2.03*** |
| | (1.28, 3.37) | (1.27, 3.36) | (1.96, 22.32) | (2.02, 23.13) | (1.25, 2.63) | (1.38, 2.97) |
| 30–34 | 1.89*** | 1.92*** | 9.37*** | 9.8*** | 1.18 | 1.52** |
| | (1.15, 3.04) | (1.16, 3.13) | (3.43, 38.63) | (3.57, 40.54) | (0.81, 1.71) | (1.03, 2.24) |
| 35–39 | 1.62* | 1.71** | 8.63*** | 8.93*** | 1.3 | 2.03*** |
| | (0.98, 2.63) | (1.01, 2.83) | (3.14, 35.71) | (3.21, 37.17) | (0.89, 1.9) | (1.35, 3.03) |
| 40–44 | 1.32 | 1.42 | 9.77*** | 10.08*** | 1.12 | 1.86*** |
| | (0.79, 2.17) | (0.83, 2.41) | (3.53, 40.56) | (3.59, 42.18) | (0.75, 1.65) | (1.23, 2.83) |
| 45–49 | 1.58 | 1.73* | 9.23*** | 9.54*** | 1.02 | 1.89*** |
| | (0.91, 2.7) | (0.96, 3.08) | (3.29, 38.56) | (3.33, 40.32) | (0.68, 1.53) | (1.22, 2.92) |
| **Education** | | | | | | |
| Incomplete Primary | | 1.02 | | 0.99 | | 1.52*** |
| | | (0.76, 1.36) | | (0.75, 1.31) | | (1.25, 1.86) |
| Complete Primary | | 1.26 | | 1.29 | | 2.07*** |
| | | (0.76, 2.17) | | (0.78, 2.07) | | (1.44, 3.01) |
| Incomplete Secondary | | 1.06 | | 1.07 | | 2.75*** |
| | | (0.74, 1.54) | | (0.74, 1.54) | | (2.11, 3.6) |
| Complete Secondary | | 1.77** | | 0.82 | | 6.38*** |
| | | (1.02, 3.22) | | (0.48, 1.36) | | (4.11, 10.25) |
| Higher | | 1.66 | | 0.77 | | 3.77*** |
| | | (0.68, 4.98) | | (0.32, 1.65) | | (1.98, 7.86) |
| **Currently Employed** | | | | | | |
| Yes | 0.98 | 0.98 | 1.01 | 1.01 | 1.19* | 1.19* |
| | (0.76, 1.27) | (0.76, 1.26) | (0.79, 1.3) | (0.79, 1.31) | (1, 1.42) | (0.99, 1.42) |
| **Age of First Cohabitation** | | | | | | |
| 10 to 14 | 1.24 | 1.27 | 0.41 | 0.41 | 0.34 | 0.4 |
| | (0.06, 8.07) | (0.06, 8.19) | (0.09, 2.17) | (0.09, 2.16) | (0.02, 2.15) | (0.02, 2.45) |
| 15 to 19 | 1.28 | 1.28 | 0.28* | 0.28* | 0.41 | 0.42 |
| | (0.07, 8.14) | (0.07, 8.04) | (0.06, 1.45) | (0.06, 1.44) | (0.02, 2.54) | (0.02, 2.52) |
| 20 or above | 1.62 | 1.52 | 0.26* | 0.26* | 0.47 | 0.4 |
| | (0.08, 10.42) | (0.08, 9.68) | (0.06, 1.34) | (0.06, 1.37) | (0.02, 2.92) | (0.02, 2.4) |
| **Polygamous Marriage** | | | | | | |
| Polygamous | 1.33 | 1.32 | 3.01*** | 2.98*** | 1.07 | 1.04 |
| | (0.65, 3.23) | (0.64, 3.2) | (1.79, 4.91) | (1.77, 4.86) | (0.64, 1.84) | (0.62, 1.79) |
| Does Not Know | 0.74 | 0.74 | 2.25* | 2.26* | 0.59 | 0.6 |
| | (0.27, 2.6) | (0.27, 2.59) | (0.79, 5.53) | (0.8, 5.55) | (0.25, 1.43) | (0.25, 1.47) |
| **Province** | | | | | | |
| Gandaki | 1.59* | 1.56* | 1.28 | 1.27 | 1.35* | 1.19 |
| | (1.01, 2.53) | (0.99, 2.5) | (0.87, 1.88) | (0.86, 1.87) | (0.99, 1.84) | (0.87, 1.63) |
| Karnali | 0.97 | 0.94 | 0.81 | 0.81 | 1.49** | 1.30* |
| | (0.63, 1.48) | (0.61, 1.43) | (0.51, 1.26) | (0.52, 1.27) | (1.09, 2.02) | (0.95, 1.78) |
| Koshi | 1.2 | 1.2 | 0.96 | 0.93 | 1.73*** | 1.59*** |

*(Continued)*

**Table 3.** (Continued)

| | Beliefs about violence against women | | Decision-making | | Control over sexuality and sex | |
|---|---|---|---|---|---|---|
| | cOR (95% CI) | aOR (95% CI) | cOR (95% CI) | aOR (95% CI) | cOR (95% CI) | aOR (95% CI) |
| | (0.79, 1.82) | (0.79, 1.83) | (0.64, 1.43) | (0.62, 1.39) | (1.27, 2.37) | (1.16, 2.19) |
| Lumbini | 1.1 | 1.11 | 0.94 | 0.94 | 1.36** | 1.34* |
| | (0.73, 1.65) | (0.74, 1.67) | (0.63, 1.4) | (0.63, 1.39) | (1.01, 1.83) | (0.99, 1.81) |
| Madhesh | 1.56** | 1.58** | 0.65** | 0.65** | 0.47*** | 0.51*** |
| | (1.01, 2.4) | (1.02, 2.45) | (0.43, 0.99) | (0.42, 1) | (0.35, 0.63) | (0.38, 0.68) |
| Sudurpashchim | 1.74** | 1.73** | 0.83 | 0.84 | 2.42*** | 2.36*** |
| | (1.09, 2.81) | (1.08, 2.8) | (0.54, 1.28) | (0.54, 1.29) | (1.74, 3.38) | (1.69, 3.31) |
| **Urban vs Rural** | | | | | | |
| Rural | 1.18 | 1.17 | 0.89 | 0.9 | 0.99 | 0.95 |
| | (0.94, 1.48) | (0.93, 1.47) | (0.71, 1.12) | (0.72, 1.12) | (0.84, 1.16) | (0.81, 1.12) |
| **Religion** | | | | | | |
| Buddhist | 1.01 | 1.03 | 1.48* | 1.47* | 1.38* | 1.46** |
| | (0.64, 1.66) | (0.65, 1.7) | (0.98, 2.2) | (0.97, 2.19) | (0.98, 1.98) | (1.03, 2.11) |
| Muslim | 0.73 | 0.76 | 1.49 | 1.51 | 0.39*** | 0.54*** |
| | (0.43, 1.29) | (0.45, 1.36) | (0.84, 2.53) | (0.84, 2.58) | (0.26, 0.58) | (0.36, 0.81) |
| Kirat | 0.95 | 0.96 | 1.28 | 1.28 | 1.89** | 1.84** |
| | (0.5, 1.93) | (0.51, 1.95) | (0.67, 2.32) | (0.67, 2.32) | (1.09, 3.47) | (1.06, 3.4) |
| Christian | 1.4 | 1.44 | 0.98 | 0.98 | 0.77 | 0.8 |
| | (0.75, 2.91) | (0.77, 2.99) | (0.51, 1.72) | (0.51, 1.74) | (0.51, 1.17) | (0.53, 1.21) |
| **Wealth Index** | | | | | | |
| Poorer | 0.91 | 0.89 | 1.11 | 1.11 | 1.41*** | 1.29** |
| | (0.67, 1.25) | (0.65, 1.22) | (0.8, 1.54) | (0.8, 1.54) | (1.13, 1.77) | (1.03, 1.61) |
| Middle | 1.02 | 0.99 | 1.28 | 1.28 | 1.88*** | 1.58*** |
| | (0.73, 1.42) | (0.71, 1.39) | (0.92, 1.77) | (0.92, 1.78) | (1.49, 2.39) | (1.24, 2.01) |
| Richer | 1.29 | 1.22 | 1.16 | 1.16 | 2.19*** | 1.57*** |
| | (0.9, 1.87) | (0.84, 1.78) | (0.82, 1.63) | (0.81, 1.65) | (1.7, 2.83) | (1.21, 2.05) |
| Richest | 1.63** | 1.39 | 0.89 | 0.93 | 3.52*** | 1.91*** |
| | (1.04, 2.61) | (0.86, 2.29) | (0.58, 1.36) | (0.59, 1.47) | (2.54, 4.94) | (1.34, 2.73) |
| **Access to Media** | | | | | | |
| Less Than Once A Week | 1.23 | 1.2 | 0.67*** | 0.67*** | 1.94*** | 1.72*** |
| | (0.93, 1.63) | (0.9, 1.59) | (0.51, 0.87) | (0.51, 0.87) | (1.6, 2.34) | (1.42, 2.08) |
| At Least Once A Week | 1.02 | 0.99 | 0.63*** | 0.63*** | 1.68*** | 1.45*** |
| | (0.76, 1.37) | (0.74, 0.290) | (0.47, 0.85) | (0.47, 0.85) | (1.37, 2.05) | (1.18, 1.78) |
| **Partner Controlling Behavior** | | | | | | |
| At Least One Type | 0.54*** | 0.55*** | 1.34** | 1.33** | 0.79*** | 0.81** |
| | (0.43, 0.68) | (0.43, 0.69) | (1.06, 1.69) | (1.05, 1.68) | (0.67, 0.93) | (0.69, 0.96) |
| **IPV Exposure** | | | | | | |
| Yes | 0.85 | 0.85 | 1.29 | 1.3 | 0.77** | 0.8* |
| | (0.62, 1.17) | (0.62, 1.18) | (0.94, 1.75) | (0.94, 1.77) | (0.61, 0.97) | (0.63, 1.01) |

Note: *p<0.1

**p<0.05

***p<0.01. Two respondents who chose "Other" religion were not included in regressions for Decision-making domain because none of them was in the top quintile.

## Discussion

While DHS provides convenient standardized data for international comparison, especially, in the context of developing economies, its selection of empowerment indicators is rather limited to grasp capture contextual peculiarities [12]. This limitation is critical as it may lead to an incomplete understanding of the multi-faceted nature of women's empowerment. Additionally, relying on indirect indicators of women's empowerment, such as education or access to information may be problematic in terms of its conceptual validity [13]. Such indicators often fail to capture the nuanced and dynamic aspects of empowerment that are deeply rooted in cultural, social, and economic contexts.

Instead, we followed a more direct approach which combines women's actions or beliefs into three domains related to empowerment and then identifies factors that improve or worsen women's empowerment [2]. This approach allows for a more nuanced and accurate depiction of women's empowerment by directly addressing their lived experiences and perspectives. It also enables a comparison of our findings for Nepal based on DHS 2022 survey with those for Mozambique based on DHS 2015 survey [2], thus, contributing to the growing literature on women empowerment.

Hence, it is of interest to compare the results in this paper based on DHS survey for Nepal for 2022 with those for Mozambique based on the 2015 DHS survey. This comparison not only underscores the importance of contextual factors but also suggests the potential evolution of empowerment dynamics over time. Age, education, polygamous marriage, region, religion, wealth index, access to media and partner controlling behavior have been significant in at least one domain of women empowerment in both studies. However, employment status, age of first cohabitation, rural status and IPV exposure were significant in some domains of women empowerment in Mozambique in 2015 but in no domains in Nepal 2022. This fact potentially reflects regional differences in sources of women empowerment but also potential changes over time, given a 7-year period that lapsed between the two surveys. These findings emphasize the need for tailored approaches in policy-making and intervention design, taking into account both regional specificities and temporal changes.

Although the paper methodologies established in previous studies and is based on a large sample of women of representative age, it is not without limitations. First, we do not observe women empowerment directly but rather construct it from women's beliefs and actions. This approach, while insightful, can be limited by the subjective nature of self-reported data. Second, while our dataset uses a rich list of women's characteristics it may still lack some essential factors that may need to be collected, such as mother's empowerment or political representation. Hence, while our results represent an important first step in better understanding of women empowerment in Nepal further research is needed when better quality data become available.

In comparing our findings with previous literature, particularly studies conducted in other regions such as Mozambique, we uncover both similarities and differences. While age, education level, wealth, access to media, and partner controlling behavior emerge as significant determinants of women's empowerment in both contexts, there are notable variations in the impact of certain factors. For example, while polygamous marriage shows a positive association with empowerment in decision-making in our study, this association was not observed in the Mozambique study. Such differences highlight the importance of contextual factors and regional dynamics in shaping women's empowerment trajectories.

It is also important to remember that when we consider programs for women's empowerment, we should do so with regard to the subtleties and regional differences of the issue. The notable regional differences found in our research report further support this by illustrating

how greater levels of empowerment have been demonstrated in places such as Madhesh and Sudurpashchim, thereby underlining the significance of contextually aligned interventions at the grassroots level.

When crafting strategies to enhance women's status and opportunities in Nepal, regional variations must be carefully considered to maximize effectiveness and long-term sustainability. Engaging local leaders and community members in the design and execution of these programs is essential to ensure they resonate with and are embraced by Nepal's diverse cultural and religious groups. This collaborative approach helps tailor initiatives to specific needs and contexts, fostering greater acceptance and impact across the country's varied landscapes and communities.

Given certain provincial differences in women empowerment, it is important to consider potential policies and interventions that policy makers (in particular, social and welfare practitioners) should seek to implement. While Madhesh province shows greater empowerment in beliefs about violence against women it falls behind the other two indexes. When it comes to the domain of decision making previous study on Sierra Leone identifies the importance of combining complementary programs on gender training with economic interventions [14]. Similarly, the base province of Bagmati falls behind 5 other regions in terms of control over sexuality and sex. Hence, policy-makers in Bagmati and Madhesh should consider the role of sex education as a critical means for women empowerment in health through increasing women's knowledge and related behaviors [15].

In addition to that, our investigation also proves that women's empowerment is a very complicated issue which consists of ideas of violence, the freedom of decisions, preferences in sex, and so on. In other words, any programs on this subject must reflect the all-rounded nature of these components with mutual interrelation among them. Addressing those social norms and structural barriers can lead to a great positive impact and drive towards gender equality in Nepal. Programs should also include measures to monitor and evaluate their impact regularly, ensuring that they adapt to the changing needs and conditions of the communities they serve.

We should also be aware of the drawbacks of this study. Despite the 2022 DHS dataset being a valuable resource for assessing women's empowerment in Nepal, our study utilizes proximate indicators that may not encompass its full magnitude. Moreover, using cross-sectional data alone will not be able to show causality and sequentiality patterns. Longitudinal studies could be used in future research to investigate the trend of women's empowerment over time as well as assess the impact and effectiveness of intervention programs towards women's empowerment.

To wrap up, the research is instrumental in adding to the body of knowledge that outlines factors influencing women's empowerment in Nepal and highlights the significance of area-based intervention policies. To summarize, our research has enhanced the understanding of the influences that determine women's empowerment in Nepal and points out the need for location-specific intervention programs. The reason why our results would be useful for such policy measures is that they identify the most influential factors affecting women's empowerment across areas of activity and can significantly contribute to equalizing men's and women's rights in Nepal. Eventually, women's empowerment should be treated not only as an issue of justice but also as a crucial element facilitating sustainable development and inclusive growth. The insights gained from this study can inform the design of targeted policies that not only address immediate empowerment needs but also foster long-term gender equality and social transformation.

## Conclusion

In this study we utilized the 2022 DHS survey for Nepal to employ Principal Component Analysis (PCA) in order to identify significant components for women's empowerment domains:

Beliefs about violence against women, Decision-making, and Control over sexuality and safe sex. We found that older age is associated with higher empowerment across all domains. Interestingly, while partner controlling behavior generally reduces empowerment in beliefs about violence and control over sexuality, it is linked to increased decision-making empowerment. Regional differences are noted, with higher empowerment in Madhesh and Sudurpashchim. Education level and wealth correlate with increased empowerment in control over sexuality and safe sex, but not in other two domains of empowerment. Access to media has mixed effects, reducing empowerment in decision-making but enhancing it in control over sexuality and safe sex. We identified many similarities but also some differences with a previous study for Mozambique which signifies the importance of regular studies of women empowerment given its potential for change over time as well as prevailing differences across regions.

In summary, our study based on 2022 DHS survey data provides important insights into the status and determinants of women's empowerment in Nepal. By applying principal component analysis (PCA) and logistic regression, we identified key factors associated with women's empowerment in three areas: beliefs about violence against women, decision-making and control over sexual behavior, and safe sex.

Our findings highlight the importance of age, education level, wealth, access to media, partner controlling behavior, and regional differences in shaping women's empowerment trajectories in Nepal. Notably, aging has been a consistent predictor of increased empowerment across all domains, highlighting the potential role of intergenerational changes in gender equality attitudes and norms.

In addition, our research underscores the importance of implementing specific measures to promote women's empowerment, taking into account the various dimensions and considering the distinct socio-economic and cultural circumstances found in different areas of Nepal. By crafting interventions that directly tackle local obstacles and capitalize on local strengths, policymakers and development practitioners can enhance the impact and long-term viability of initiatives aimed at empowering women.

We must recognize the constraints of our study, which include relying on indirect measures of empowerment and the cross-sectional nature of our data. To obtain a more thorough understanding of the ever-evolving process of women's empowerment in Nepal, future studies should employ longitudinal methodologies and qualitative approaches.

To sum up, in the promotion of women's empowerment, there is not only a moral duty but also a strategic investment to accomplish sustainable development and inclusive growth. Addressing both structural barriers and social norms that contribute to gender inequality will allow Nepal to unleash its female potential and lead progress towards the Sustainable Development Goals (SDGs) as well as the creation of an equal society.

## Acknowledgments

We are grateful to the Demographic and Health Survey Program for the opportunity to use 2022 dataset for Nepal for this study.

## Author Contributions

**Conceptualization:** Daan-Max van Dongen.

**Data curation:** Vladyslav Shymanskyi.

**Formal analysis:** Vladyslav Shymanskyi.

**Investigation:** Vladyslav Shymanskyi.

**Methodology:** Daan-Max van Dongen, Maksym Obrizan.

**Project administration:** Daan-Max van Dongen.

**Software:** Vladyslav Shymanskyi.

**Supervision:** Daan-Max van Dongen.

**Validation:** Daan-Max van Dongen, Maksym Obrizan.

**Writing – original draft:** Daan-Max van Dongen, Maksym Obrizan, Vladyslav Shymanskyi.

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
