## [Decision Letter · Decision Letter 0]

16 Jul 2024

PONE-D-24-12705Determinants of women's empowerment in NepalPLOS ONE

Dear Dr. Obrizan,

Thank you for submitting your manuscript to PLOS ONE. After careful consideration, we feel that it has merit but does not fully meet PLOS ONE’s publication criteria as it currently stands. Therefore, we invite you to submit a revised version of the manuscript that addresses the points raised during the review process.

We look forward to receiving your revised manuscript.

Kind regards,

Kuo-Cherh Huang

Academic Editor

PLOS ONE

Journal Requirements:

4. We note you have included a table to which you do not refer in the text of your manuscript. Please ensure that you refer to Table 3 in your text; if accepted, production will need this reference to link the reader to the Table.

**Additional Editor Comments:**

Dear Dr. Obrizan,

We appreciate your submission to PLoS ONE. Although your paper is interesting, the reviewer had provided a variety of important concerns and helpful suggestions. Please respond to each comment of the reviewer carefully and thoroughly. In particular, please pay attention to those fundamental statistical issues raised by the reviewer. Moreover, please enrich the Discussion section. Please explain where you feel you cannot completely agree with reviewers’ suggestions, if any.

Thank you.

Kuo-Cherh Huang

Academic Editor

Reviewers' comments:

Reviewer's Responses to Questions

**Comments to the Author**

1. Is the manuscript technically sound, and do the data support the conclusions?

Reviewer #1: Yes

2. Has the statistical analysis been performed appropriately and rigorously? 

Reviewer #1: Yes

3. Have the authors made all data underlying the findings in their manuscript fully available?

Reviewer #1: Yes

4. Is the manuscript presented in an intelligible fashion and written in standard English?

Reviewer #1: Yes

5. Review Comments to the Author

Reviewer #1: Review PONE-D-24-12705

Dear authors, thank you for the opportunity to review your manuscript titled, “Determinants of women's empowerment in Nepal.” Find a few comments below:

− Please explain why you categorize the age variable. While there are intervention reasons (as alluded to in the discussion section), statistically information is lost when you categorize. I am not asking you to re-run your data with the continuous age variable but do provide your reader with a justification.

− PCA uses continuous variables, please explain how you used categorical variables. And to clarify, it is the variables in Table 1 that were used in the PCA to generate the empowerment index—correct? Please clarify

− Reference Table 3 somewhere within the text. Are these results from the logistic regression models?

− The provincial breakdown in the bivariate analysis is very much appreciated. Please tell us a little bit about the geo-political structure of these provinces to help us contextualize your findings.

− I was also expecting a little more critical analysis in the discussion section more so given the compelling regional comparisons presented in the results section. However, I find this section rather generic. For instance, what policies/programs/interventions stand to benefit from these results? What can health/social welfare practitioners glean from these results?

All the best!

6. PLOS authors have the option to publish the peer review history of their article (what does this mean?). If published, this will include your full peer review and any attached files.

Reviewer #1: No

---

## [Author Response · Author response to Decision Letter 0]

15 Aug 2024

Point-by-point responses provided in a separate document.

---

## [Editor Report · Decision Letter 1]

27 Aug 2024

Determinants of women's empowerment in Nepal

PONE-D-24-12705R1

Dear Dr. Obrizan,

We’re pleased to inform you that your manuscript has been judged scientifically suitable for publication and will be formally accepted for publication once it meets all outstanding technical requirements.

Kind regards,

Kuo-Cherh Huang

Academic Editor

PLOS ONE
---

## [Editor Report · Acceptance letter]

3 Sep 2024

PONE-D-24-12705R1 

PLOS ONE

Dear Dr. Obrizan, 

I'm pleased to inform you that your manuscript has been deemed suitable for publication in PLOS ONE. Congratulations! Your manuscript is now being handed over to our production team.

Kind regards, 

on behalf of

Dr. Kuo-Cherh Huang 

Academic Editor

PLOS ONE